# A hybrid security system for drones based on ICMetric technology

**Khattab M. Ali Alheeti**[1]***, Fawaz Khaled Alarfaj**[2]***, Mohammed Alreshoodi**[3]**,
**Naif Almusallam**[2]**, Duaa Al Dosary**[1]

**1** Computer Science Department, College of Computer Sciences & Information Technology, University of Anbar, Ramadi, Iraq, **2** King Faisal University, Hufof, Saudi Arabia, **3** Unit of Scientific Research, Applied College, Qassim University, Buraydah, Saudi Arabia

* co.khattab.alheeti@uoanbar.edu.iq (KMAA); falarfaj@kfu.edu.sa (FKA)

**Data Availability Statement:** In this study, the dataset represents the Wi-Fi traffic data records used. Two types of data exist within this set: the bidirectional flow mode that compromised 54

## Abstract

Recently, the number of drones has increased, and drones' illegal and malicious use has become prevalent. The dangerous and wasteful effects are substantial, and the probability of attacks is very high. Therefore, an anomaly detection and protection system are needed. This paper aims to design and implement an intelligent anomaly detection system for the security of unmanned aerial vehicles (UAVs)/drones. The proposed system is heavily based on utilizing ICMetric technology to exploit low-level device features for detection. This technology extracts the accelerometer and gyroscope sensors' bias to create a unique number known as the ICMetric number. Hence, ICMetric numbers represent additional features integrated into the dataset used to detect drones. This study performs the classification using a deep neural network (DNN). The experimental results prove that the proposed system achieves high levels of detection and performance metrics.

## Introduction

Unmanned aerial vehicles (UAVs), commonly called drones, have become more readily available, more common, and more sophisticated. UAVs support new capabilities, such as autonomous behaviour and increased data collection. Currently, due to their ease of use and accessibility, they are exploited for commercial purposes in various fields, such as surveying, delivering medicines and goods, public surveillance, agriculture, cartography, and first aid. The primary types of UAVs are presented in Fig 1.

The rapid and comprehensive development of UAVs has attracted the attention of researchers and prompted them to propose further research in the domains of defence/protection against attacks. Cybersecurity attacks can involve hacking, hijacking, or loss of the communication signals of the drone. It is also challenging to ensure the safety of UAVs moving in unwanted areas, which produces weak robustness against various cybersecurity attacks [1]. Due to the UAV network connection, it is easy for adversaries to target or control. Hence, an intelligent anomaly detection system for drones is required to ensure their safety and provide identification. In this paper, the identification and security of drones were achieved and ensured. The proposed approach is based on a new technology called an integrated circuit

features and the unidirectional flow mode that compromised 18 features.

**Funding:** This work was supported by the Deanship of Scientific Research, Vice Presidency for Graduate Studies and Scientific Research, King Faisal University, Saudi Arabia [Grant No. 2976].

**Competing interests:** The authors have declared that no competing interests exist.

metric (ICMetric), which heavily depends on each device's internal behavior. ICMetric is a new technique based on the concept of feature extraction for digital devices. However, it depends on devices' special features to generate a single ICMetric number because every device is singular in its internal environment [2, 3] Microelectromechanical system (MEMS) sensors are utilized to generate ICMetric numbers. In other words, the bias readings extracted from these sensors will create this unique security number. The main contributions of this paper are summarized below:

1. ICMetric technology is employed for the first time for drone security.

2. The use of ICMetric technology aims to generate single/unique numbers by considering additional features for the detection process.

3. A deep neural network is used as a classifier during the training phase for the detection system.

The rest of this paper is organized as follows: Section 2 presents related works. Section 3 presents the details of integrated circuit metric technology. Section 4 presents the methodology of the proposed system. Section 5 evaluates the efficiency of this approach through experimental results. Section 6 presents the discussions, and finally, Section 7 presents the conclusions and possibilities for future research.

## Related works

This section summarises the latest research in related fields. Gritzalis et al. presented a literature review of various sensor techniques, such as accelerometers, gyroscopes, and magnetometers. These techniques are proposed for preventing, detecting, identifying, and mitigating damage

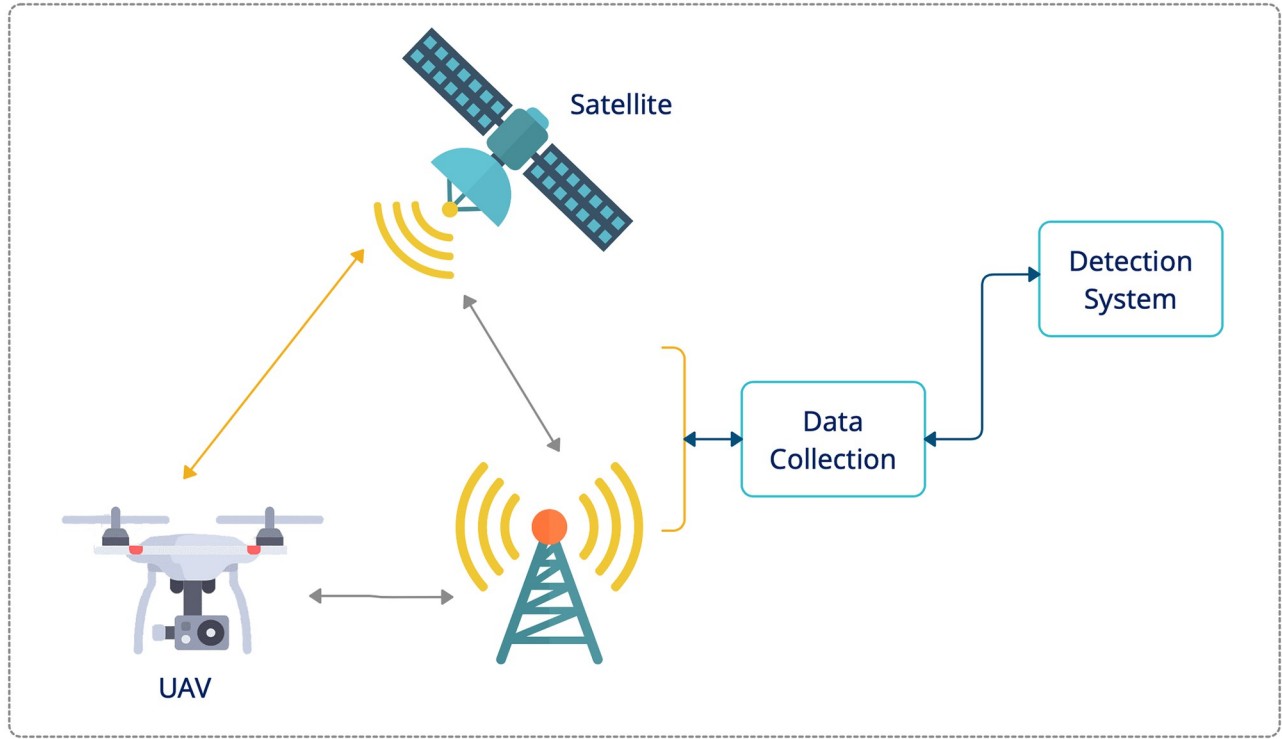

**Fig 1. Unmanned aerial vehicles (UAVs).**

from rogue drones, as well as a detailed survey of drone incidents near airports [4]. In the malicious purposes field, drones are exploited for different uses, which were reviewed in [5], along with other detection methods. The aforementioned study also analyzed the vulnerabilities of drones exploited through communication links and smart devices and hardware. Additionally, a detailed review of drone usage for various purposes in areas such as civilian, terrorist, and military domains was presented. [6] classified intrusion detection system (IDS) mechanisms used with attacks and vulnerabilities in networked drone environments. They classify the IDS mechanisms based on information gathering sources, detection methods, deployment strategies, IDS acknowledgement, detection states, and intrusion types. Another survey aimed to analyze the security threats against unmanned aircraft systems is presented in [7].

Meanwhile, [8] proposed a protection system based on a convolutional neural network (CNN) for drone detection. The system was divided into two stages, where the first aimed to detect moving objects, while the subsequent second stage intended to classify the object detected as a bird, drone, or background. The results proved that the proposed method could achieve accuracy comparable to other methods. In [9], deep learning techniques were utilized to propose an object detection system for drone detection. Various CNN techniques were used in the detection system to evaluate and test the dataset. The results showed that the proposed method based on one of the CNN techniques presented better performance than others with respect to the training dataset. Other research, based on CNN, for presenting an algorithm to solve the problem of drone detection was proposed in [10], which yielded high recall and precision values.

In [11], the authors presented a new system for detecting and tracking UAVs through a single camera mounted on various UAVs. This approach can detect and track small UAVs efficiently with limited computing resources. Moreover, they propose a lighter, inexpensive, safer, and miniaturized radar system [11]. Different types of anomalies were accurately detected by the intrusion detection system proposed for drone communication networks in [12]. An adaptive intrusion detection system for drone identification based on deep learning techniques was presented in [1]. They maintain the high true-positive rate of the IDS by using a support vector machine (SVM). The results showed that the proposed IDS could be deployed against cybersecurity attacks on UAVs with high accuracy, specificity, and sensitivity levels.

The increased frequency of attacks against drones requires an intelligent intrusion detection system for identification and protection. In [13], a novel model for intrusion detection in drones based on one-class classifiers was proposed. The model was effective across multiple UAV platforms with platform-specific F1 scores of 99.56% and 99.73% for benign and malicious sensor readings. In [14], a framework for identifying malicious attacks on the sensors of small, unmanned aircraft systems was presented. They illustrate the proposed model by detecting a spoofing attack on the global positioning system (GPS). Different conclusions were drawn using this model with a simulation dataset. [15] presented a framework for attack detection that could be used to protect the environment of medical delivery drones with active capability, delivery reliability, and other medical applications.

Having analyzed systems proposed in previous studies, this study can be distinguished from others in that it presents an identification/detection system based on the ICMetric technology that depends on individual devices' internal behavior to provide a unique number for identification. The proposed approach has been developed to train the proposed security system by using the deep neural network technique.

## Methodology

This section presents the method used to perform the proposed ICMetric detection system.

## Dataset description

The consumer UAV market has developed substantially over recent years. Regardless of its enormous potential for economic development through supporting different applications, the advancement of shopper UAVs presents expected dangers to public security and individual protection. It is critically necessary to distinguish and recognize attacking UAVs to limit these dangers proficiently. Given that purchaser UAVs are usually utilized in a regular civilian environment, existing identification techniques (such as radar, vision, and sound) may become ineffectual in numerous situations. UAVs' encrypted Wi-Fi traffic information records can be an exceptionally encouraging source to identify UAV intruders. In this work, Unmanned Aerial Vehicle (UAV) Intrusion Detection Datasets used. The dataset has been achieved from the source [16] and is available.

Notwithstanding precision, the expected runtime of the indicator technique is exceptionally significant considering that UAV gatecrashers fly many metres each second and acquire the forecast as quickly as expected. In this way, the precision and runtime of the forecast should be improved. Subsequently, pruning and working on the model would be exceptionally significant: for example, pruning the redundant highlight ages and blending the common calculations.

In this study, the dataset represents the Wi-Fi traffic data records used. Two types of data exist within this set: the bidirectional flow mode and its 54 features and the unidirectional flow mode with 18 features [16]. For this dataset, three types of drones have been utilized:

- Parrot Bebop 1

- DBPower UDI

- DJI Spark

These are as shown in Fig 2:

In this paper, the unidirectional flow mode was used to test/evaluate the security system's performance. Meanwhile, this dataset is composed of nine features that are explained in Table 1. The dataset is a combination of various UAVs and traffic behaviour modes. Fig 3 shows a snapshot of dataset sample.

## Integrated circuit metric technology (ICMetric)

Encryption systems depend on calculations that rely upon the utilization of the secret key. In attempting to improve success, however, expanding the size of the key generally does not

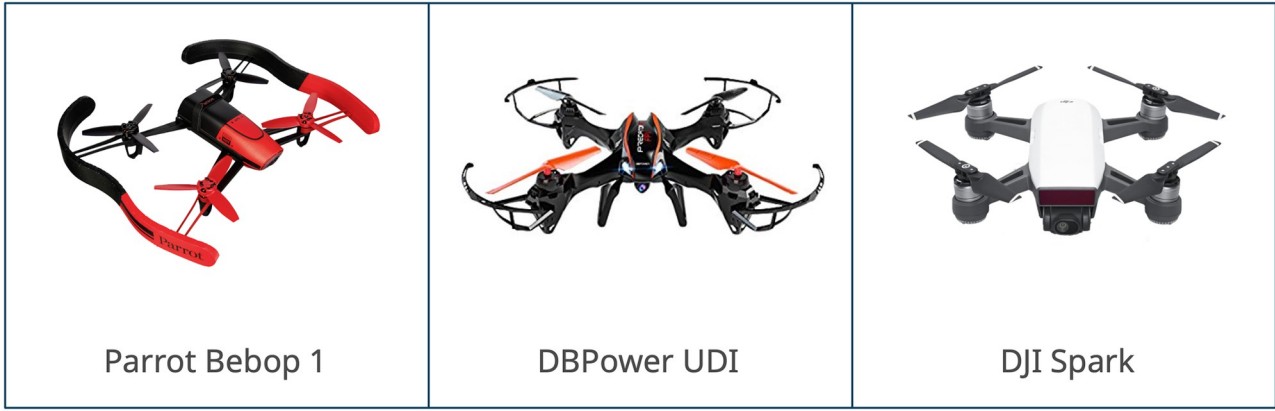

**Fig 2. Types of drones used in the dataset [16].**

**Table 1. Statistical measures and features [16].**

| Feature ID: Name | Description |
|---|---|
| $V_1$: mean | $\bar{x} = \frac{1}{N} \sum_{i=1}^{N} x(i)$ |
| $V_2$: median | the higher half value of a data sample |
| $V_3$: $MAD^1$ | $MAD = median(|x(i) - median(x)|)$ |
| $V_4$: $STD^1$ | $\sigma = \sqrt{\frac{1}{N-1} \sum_{i=1}^{N} (x(i) - mean(x))^2}$ |
| $V_5$: Skewness | $\gamma = \frac{1}{N} \sum_{i=1}^{N} (x(i) - mean(x)/\sigma)^3$ |
| $V_6$: Kurtosis | $\beta = \frac{1}{N} \sum_{i=1}^{N} (x(i) - mean(x)/\sigma)^4$ |
| $V_7$: MAX | $H = (Max(x(i))|_{i=1...N})$ |
| $V_8$: MIN | $L = (Min(x(i))|_{i=1...N})$ |
| $V_9$: Mean Square (ms) | $MS = \frac{1}{N} \sum_{i=1}^{N} (x(i))^2$ |

ensure the security of the framework or prevent theft. The ID of the device, dependent on the exceptional features of every device, serves to prevent theft. Other equipment procedures vary with respect to the ICMetric innovation in the choice of device attributes. Conventional finger-printing strategies rely on qualities that are effortlessly designed to detect, misdirect or reduce the damage inflicted by aggressors. ICMetrics use a portion of the properties that expand the intricacy of ICMetric and are difficult for an attacker to predict. Different features with uncommon application use are employed, such as perusing narratives, camera goals, normal client documents, and framework profiles [17].

ICMetrics focuses on another strategy that can utilize the features from the equipment and programming climate of a framework. Each device is unique in its internal climate; thus, the features that make each device different can be utilized to generate a single, one-of-a-kind number for every device. This depends on the following ideas [3].

1. The number is not stored in the framework and can be reproduced when required.

2. If the system is assaulted, there will be no burglary since the ICMetric is not stored.

3. The number and any procedure results that depend on the ICMetric number will be changed if any change to the product, equipment or climate is enacted.

4. There is no need for storage in any format that can serve the function of device approval.

ICMetric is a technology that has advanced alongside the security requirements for digital devices. It exploit characteristics of the model to current security. This represents a new

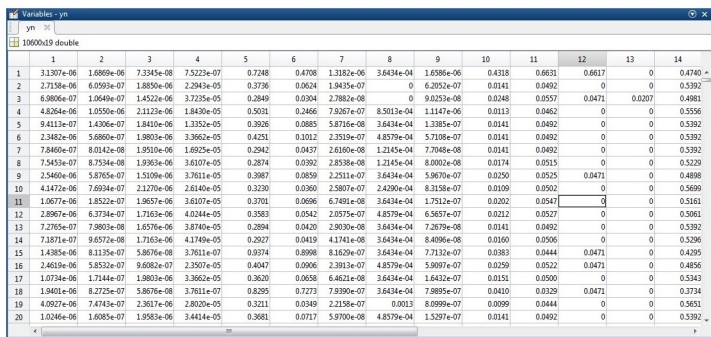

**Fig 3. Snapshot of dataset sample.**

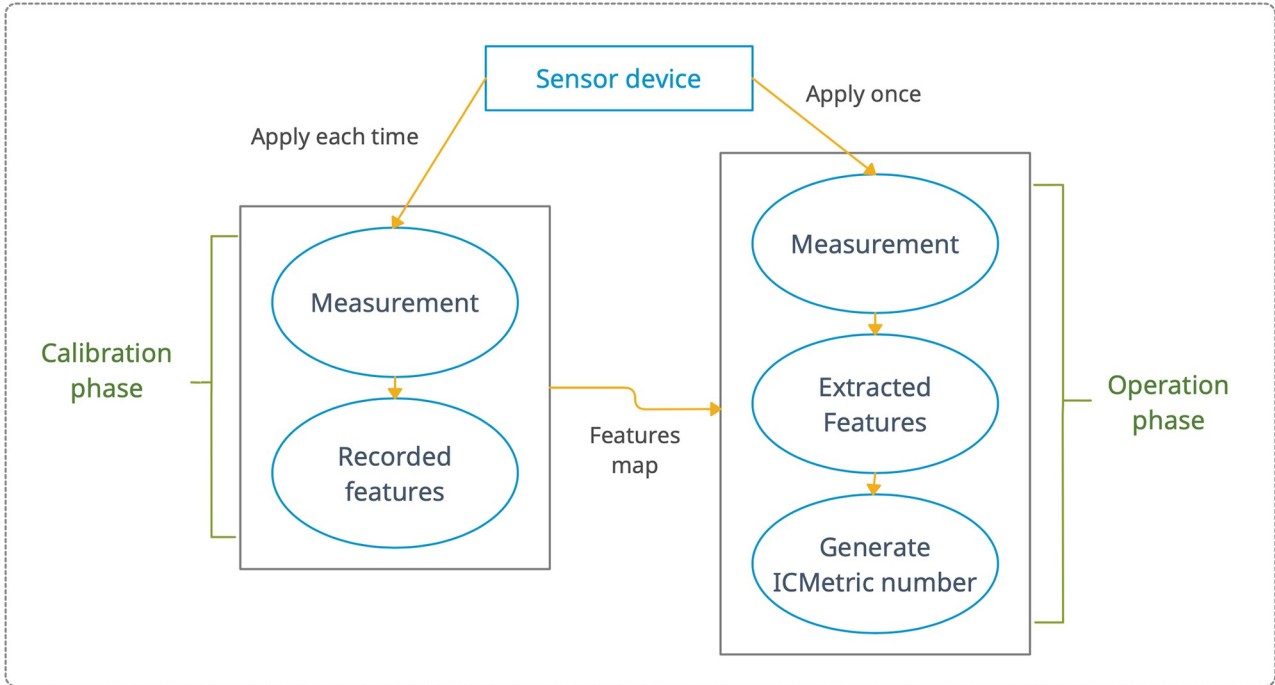

**Fig 4. Flow diagrams of ICMetric phases [19].**

technology that can extract software and hardware attributes. Each device is unique in its internal environment. Therefore, the features that make each device different from another can create an impressive number of known ICMetric numbers. These numbers are not saved and can be regenerated when required. As these numbers are not saved/stored, there will be no theft if the system is exposed to attacks [3]. The two phases of ICMetric are [18]:

**Calibration Phase**: This involves measurement of the desired feature values; generation of feature distributions for each feature illustrating the frequency of each occurrence of each discrete value for each sample device; and normalization of the feature distributions and generating normalization maps for each feature.

**Operation Phase**: This involves generating the ICMetric number by exploiting the device's internal behaviour; this number is uniquely used as an additional feature for detection.

As discussed above, the generation of an ICMetric system requires two phases, which are presented in Fig 4.

## ICMetric number generation

ICMetric generation starts once a device's features have been extracted. Singular features of devices were collected, and then statistical processes were provided. The creation of ICMetric required two phases: a calibration phase and an operational phase. These phases were applied as and when required, after which the ICMetric and any relevant information were disposed of. The ICMetric and related component information is rarely imparted during any period of creation or use. This system uses ICMetric technology to generate a unique ICMetric number used for identification/detection, which will be integrated into the dataset used in this research as additional features for identification. The generation of this number is based on a mathematical model and requires some statistical analysis for this purpose. The generation process

requires a probability mass ($x$) function to determine the precise value from the bias reading because these readings are discrete random variables.

$$p(x) = \frac{1}{\sigma\sqrt{2n}} e^{\frac{-(x-\mu)^2}{2\sigma^2}}, \tag{1}$$

If $\bar{x}$ represents the mean, $x$ represents a specific number of readings extracted from the accelerometer and gyroscope, while $n$ represents a total number of readings [19].

$$\bar{x} = \frac{1}{n}\sum_{i=1}^{n} x_i, \tag{2}$$

For ICMetric generation, the standard deviation ($\sigma^2$) is calculated

$$\sigma^2 = \sum_{i=1}^{n} P(x_i)(x_i - \bar{X})^2, \tag{3}$$

Furthermore, variance ($s^2$) is calculated:

$$s^2 = \frac{1}{n-1}\sum_{i=1}^{n} (x_i - \bar{X})^2, \tag{4}$$

The skewness distribution ($S$) is calculated by Eq (5):

$$S = \frac{3(\bar{X} - m)}{s^2}, \tag{5}$$

The confidence interval was used as calculated by Eq (6) [2].

$$CI = \bar{X} \pm \upsilon\frac{\sigma}{\sqrt{n}}, \tag{6}$$

In this paper, a security system is proposed for anomaly detection in a drone's environment. The scheme is heavily based on readings generated by MEMS (gyroscope and accelerometer) sensors. However, the bias readings extracted from these sensors were utilized to create ICMetric numbers. An ICMetric number is a unique number that fulfilled a vital role as additional features were added to the dataset used in this work.

## The proposed ICMetric detection system

The ICMetric technique depends heavily on measurable features, which were extracted from the properties of a particular digital device. The focus was on utilizing MEMS sensors in the form of gyroscopes and accelerometers.

For this research, a drone is required, where MEMS sensors are embedded and used to provide bias readings. These bias readings are employed to create an ICMetric number that is used as identification for the drone. The steps of the ICMetric detection system are shown in Fig 5.

After dataset selection, the ICMetric number was generated by a simple mathematical model as mentioned. To generate an ICMetric number, two types of MEMS sensors are utilized: the accelerometer and gyroscope. Furthermore, bias readings were extracted from these sensors to generate ICMetric numbers used as additional features integrated into the drone's detection dataset. Two phases, the training and testing phases, were utilized to evaluate the

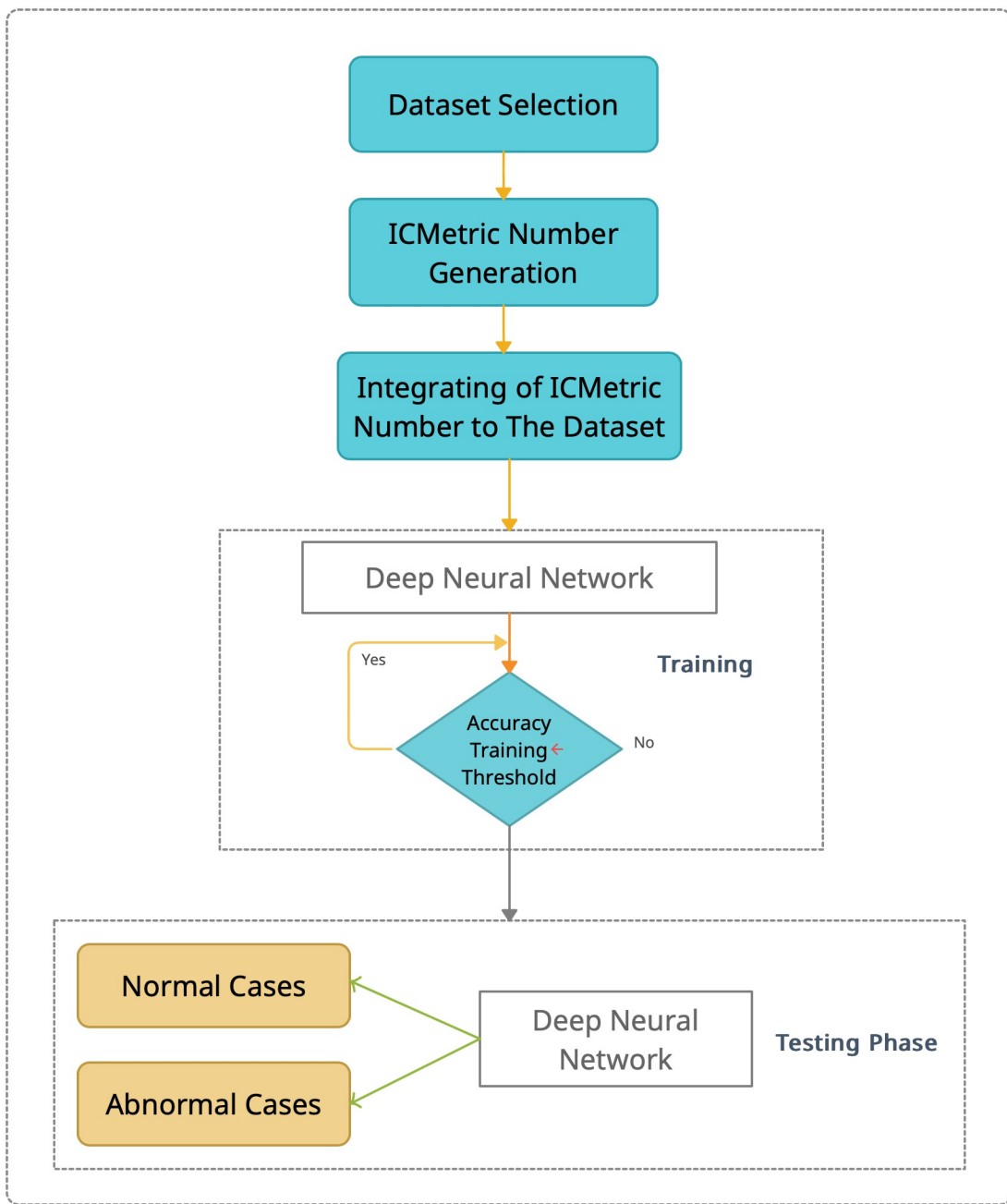

**Fig 5. ICMetric system block diagram.**

proposed system's efficiency. Hence, DNN is being used as an intelligent classifier for system evaluation.

## Deep neural network

Deep neural network (DNN) was used to train the dataset for the proposed anomaly detection system. DNN is a subfield of machine learning used in this study as a detection classifier. More specifically, a DNN is a classification learning algorithm that uses a machine learning concept

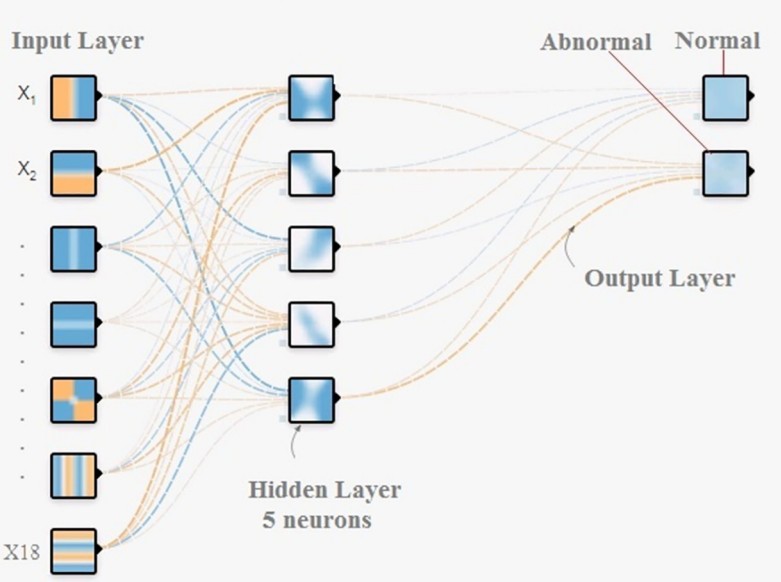

**Fig 6. DNN basic structure.**

to maximize the accuracy of prediction. Moreover, DNNs are practical, robust, and accurate and are becoming increasingly important when operating in the training and testing phases.

Here, a DNN is employed to train the dataset for the proposed detection system. The DNN has three main layers: an input layer that consists of 18 neurons, three hidden layers, and an output layer. This proposal relies on increasing the detection system accuracy and decreasing the false alarm rate. Each hidden layer consists of four neurons in each hidden layer, while the output layer features two that are normal and abnormal. The testing phase was utilized to determine the system's efficiency with respect to error rate, detection accuracy rate, and four alarm types (confusion matrix).

The measurements for choosing the best design of the DNN used in the proposed model relied on trial-and-error and the best ratio principle for neural network training. In this study, the reason for designing a DNN with 18 neurons in the input layer relates to the number of features used and will be explained further in a later section. Moreover, according to the trial-and-error principle, the optimal number of neurons for designing the hidden layer is four layers with five neurons. The best DNN structure used is presented in Fig 6.

The initial parameters that have a direct impact on the classification performance of the training phase used in the DNN are presented in Table 2.

**Table 2. Initial parameters.**

| Parameter | Value |
| --- | --- |
| Train Parameter learn | $1 * 10^{-7}$ |
| Train Parameter goal | 0 |
| Gaussian Radial Basis Function | 1 |
| Train Parameter epochs | 23 |
| Train Parameter min_grad | $1 * 10^{-13}$ |
| BoxConstraint | $1^{e6}$ |

**Table 3. Accuracy rate with different numbers of layers and neurons.**

| Number of Hidden Layers | Performance | Epochs | Time | Training Accuracy |
|---|---|---|---|---|
| Two hidden layers with 5 neurons | 0.41 | 22 | $18_s$ | 46.79% |
| Three hidden layers with 5 neurons | 0.8 | 25 | $26_s$ | 52.6% |
| Four hidden layers with five neurons | 2.0099e16 | 38 | $29_s$ | 100% |
| Six hidden layers with 5 neurons | 0.373 | 11 | $31_s$ | 99.96% |

In more details, the algorithm of deep neural network is shown below:

**Algorithm 1** Deep Nural Network Algorithm

```
Input: Wi-Fi traffic data records after integrated with ICMetric.
Output: Anomaly detection results.
```
Initialize: $w_0, ..........w_n$
Iterate:
**For** $i$ = 0, 1, 2..... **do**
Choose learning rate $\eta_t$ > 0.
Choose another parameter, goal, Gaussian radial function,..... (Table 1).
Starting training anomaly detection with dataset (ICMetric).
Applying threshold (99.00%) otherwise return on training phase again.
Testing phase for detection system with others.
Applying some performance criteria metrics to measure effective of the security system.
**Result:** Normal or Attack.

## Experimental results

After dataset selection, the system is ready for training/testing phases to evaluate the performance of the proposed detection system-the total accuracy of the training reached 99.99%.

In this study, DNN is employed as an intelligent classifier to achieve more accurate results. The DNN used in this work consists of three layers: an input layer, hidden layers, and an output layer. The trial and error principle is utilised to select the number of hidden layers based on the accuracy rate. Table 3 shows the number of hidden layers that can affect the proposed detection system's performance metrics.

According to Table 3, it is readily noticeable that the optimal number of hidden layers is four, with five neurons in each layer. Fig 7 shows the training phase with an optimal number of hidden layers with neurons.

Figs 8–10 show the performance and training status of the DNN in this detection approach.

Some metrics were utilized to calculate accuracy, effectivity, confusion matrix, and recall. These were then calculated to evaluate the efficiency of the proposed system as presented

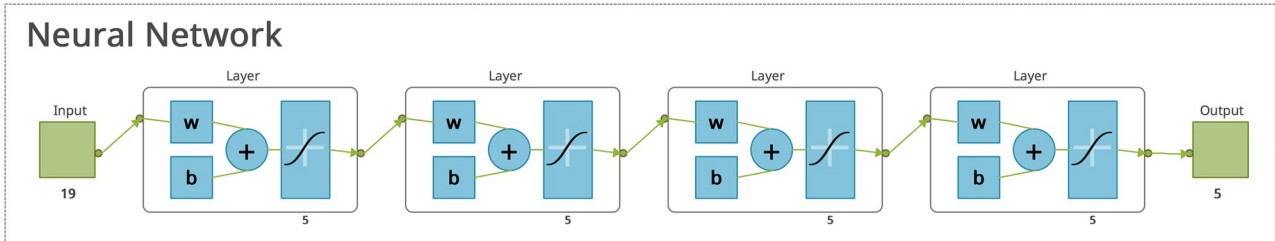

**Fig 7. Training phase with optimal number of hidden layers.**

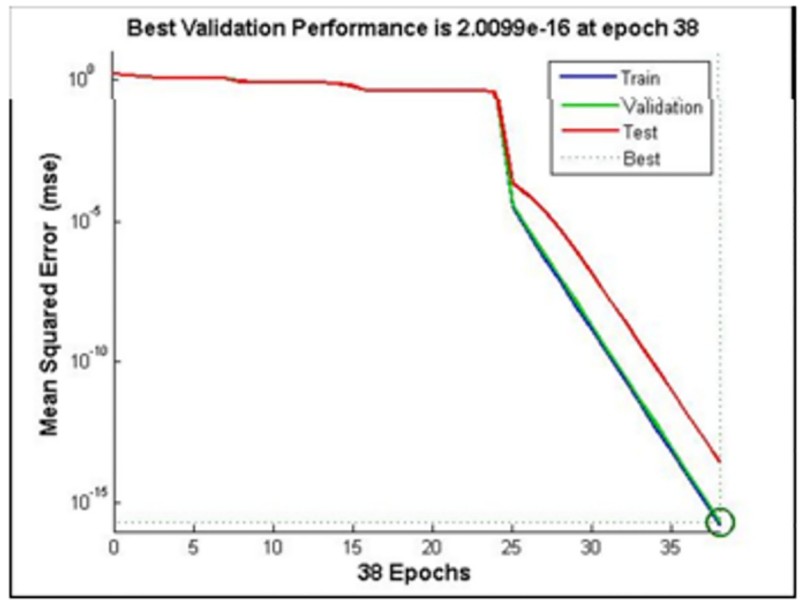

**Fig 8. Training performance of DNN—MSE.**

below [20]:

$$Accuracy = \frac{Number\ of\ correctly\ classified\ patterns}{Total\ number\ of\ patterns}, \tag{7}$$

To measure and evaluate the proposed system performance, four types of alarms are needed: true positive (TP), true negative (TN), false positive (FP), and false negative (FN).

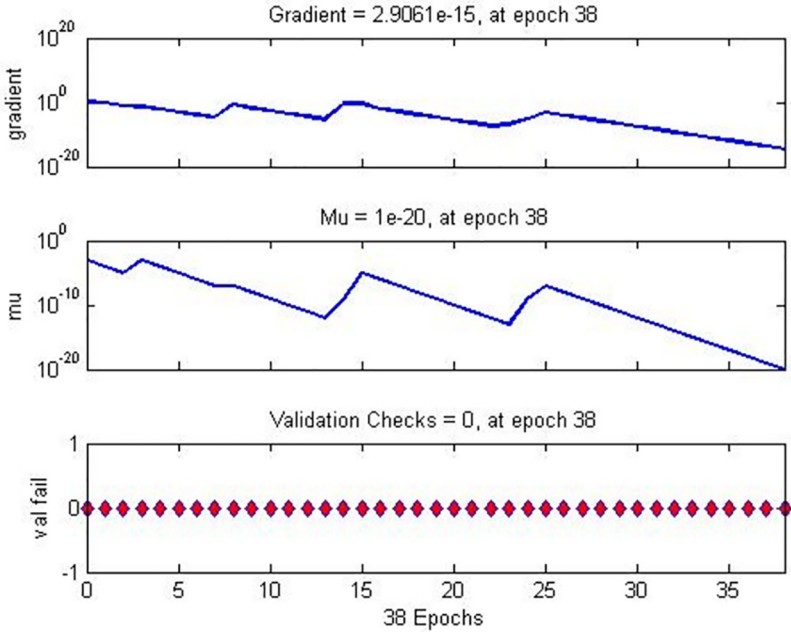

**Fig 9. Training state of DNN—Gradient.**

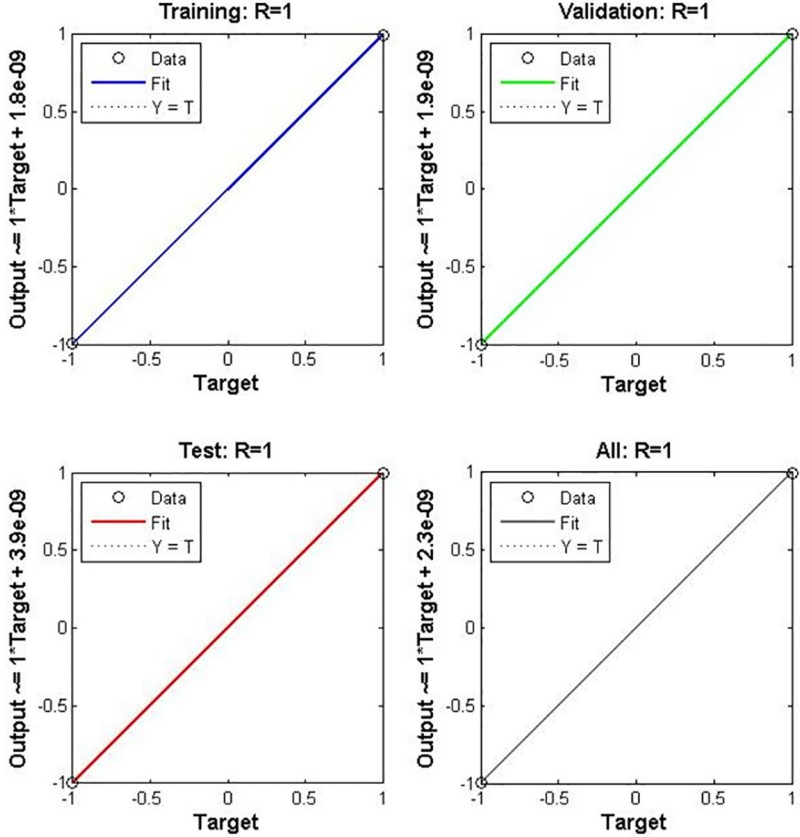

**Fig 10. Training status of DNN—Validation.**

These metrics are calculated depending on the equations in 8–11:

$$TP_{Rate} = \frac{TP}{TP + FN}, \tag{8}$$

$$TN_{Rate} = \frac{TN}{TN + FP}, \tag{9}$$

$$FN_{Rate} = \frac{FN}{FN + TP}, \tag{10}$$

$$FP_{Rate} = \frac{FP}{FP + TN}, \tag{11}$$

To evaluate efficiency, the proposed system tested for some metrics with a testing rate of 99%. Table 4 shows the classification system performance and the number of feature records used in the detection system that utilized ICMetric technology.

According to Table 4, higher rates of accuracy were achieved. Moreover, a higher rate of accuracy proved the result of using a balanced dataset achieved by a trusted source in [19]. In addition, ICMetric numbers are generated by a simple mathematical model and integrated with the dataset, resulting in a high accuracy rate. Using the equations in 8–11, the confusion matrix or alarm rate was calculated. The results are shown in Table 5.

**Table 4. Classification accuracy rate with ICMetric.**

| Round 1 with ICMetrics | | | | | |
|---|---|---|---|---|---|
| Class | Actual Dataset | Detection System | Match Record | Miss Type | Accuracy Ratio |
| Normal | 2805 | 2803 | 2803 | 0 | 99.92% |
| Abnormal | 2195 | 2195 | 2195 | 0 | 100% |
| Unknown | 0 | 2 | 0 | 2 | Nan |
| **Round 2 with ICMetrics** | | | | | |
| Class | Actual Dataset | Detection System | Match Record | Miss Type | Accuracy Ratio |
| Normal | 2805 | 2850 | 2850 | 0 | 100% |
| Abnormal | 2150 | 2149 | 2149 | 0 | 99.95% |
| Unknown | 0 | 1 | 0 | 1 | Nan |
| **Round 3 with ICMetrics** | | | | | |
| Class | Actual Dataset | Detection System | Match Record | Miss Type | Accuracy Ratio |
| Normal | 2773 | 2773 | 2773 | 0 | 100% |
| Abnormal | 2227 | 2225 | 2225 | 0 | 99.91% |
| Unknown | 0 | 2 | 0 | 2 | Nan |

**Table 5. Average alarm rate.**

| Alarms | Ratio |
|---|---|
| True Positive | 98.76% |
| True Negative | 100% |
| False Negative | 1.24% |
| False Positive | 0% |

The ratio of the correctly positively labelled results to the total number of all positive labels is known as the detection precision.

With various numbers of drones, the classification rate was calculated as presented in Table 6.

To evaluate the DNN classifier's performance, the receiver operating characteristic (ROC) curve was measured. ROC shows the performance of the metrics for classification problems in defining the threshold of a system. The true positive rate and the false positive rate are two parameters improved by plotting the ROC curve, as presented in Figs 11 and 12.

Fig 11 shows the FP rates plotted value as the decision threshold is varied to present the ROC curve.

**Table 6. Classification rate with various numbers of drones.**

| Number of Drones | Accuracy Ratio | Time |
|---|---|---|
| 4 | 94% | 0.83s |
| 5 | 91.82% | 0.84s |
| 6 | 90.27% | 2.35s |
| 7 | 87.2% | 3.82s |
| 8 | 83.61% | 4.69s |
| 9 | 82.7% | 4.83s |

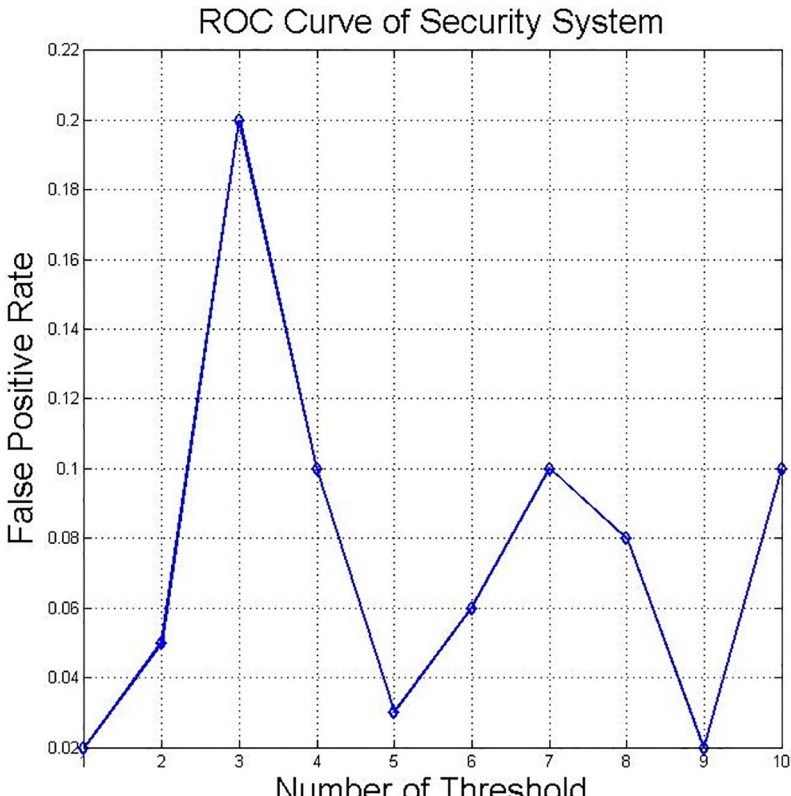

**Fig 11. ROC curve for DNN—FPR.**

Fig 12 shows the TP rate's plotted value as the decision threshold is varied to present the ROC curve.

## Discussion

As presented in Tables 4–6, the detection system that is heavily based on ICMetric technology and DNN as classifiers is more effective, fast and efficient, with low error rates in detecting abnormal drones. The proposed method has presented acceptable performance results for detecting malicious drones. In Table 6, we can notice that the proposed approach has greater malicious detection ability when the number of drones is increased. The proposed ICMetric system is compared with another ordinary system to prove that the proposed system can achieve better accuracy when drones increase, as shown in Table 7. However, we can easily notice that the proposed system's accuracy rate fluctuates because it is affected by many external factors, such as behavior, signal jamming, traffic mode, and mission type.

Fig 13 compares the proposed ICMetric system with another ordinary system (without ICMetric) to prove that the proposed system was more accurate when the number of drones increased. However, we can easily notice that the accuracy rate fluctuated because it was affected by many external factors, such as behaviour, signal jamming, traffic mode, and mission type. Finally, to demonstrate the performance of this proposed system, its performance was compared with existing IDS system UAVs. These systems exploit different specifications, models and learning algorithms.

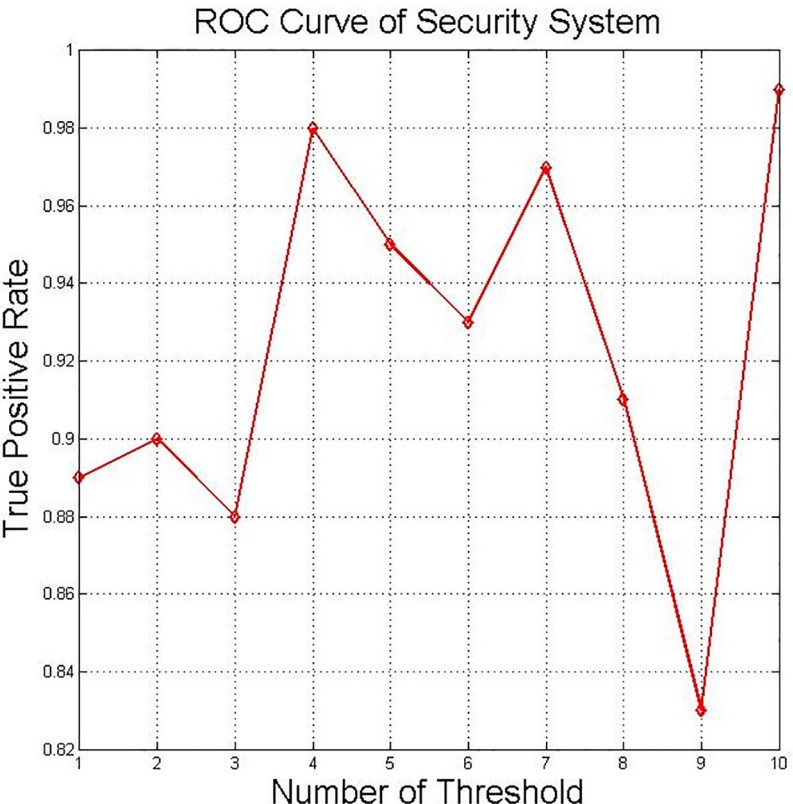

**Fig 12. ROC curve for DNN—TPR.**

In Table 7, the accuracy is utilized as a performance metric to enable comparison of the performance of the proposed system with those of different existing IDS methods. Consequently, it can be observed that our solution achieved a better accuracy rate of 99.99% compared to other IDS models for UAVs.

## Conclusions and future directions

In this paper, an intelligent anomaly detection system was proposed for drone security against attacks depending on ICMetric technology. The proposed approach was implemented in the

**Table 7. Comparison with existing IDS systems.**

| Name | Year | Accuracy | Dataset | Classifier | Security Target |
|---|---|---|---|---|---|
| [21] | 2019 | 91.50% | KDD Cup 99 | Particle Swarm Optimization (PSO) | Provides a solution to solve intrusion detection problems of UAV networks |
| [8] | 2020 | 99.83% | Drone-vs-Bird challenge dataset | CNN | Proposes a real-time drone detection algorithm to accomplish the task of drone detection |
| [22] | 2016 | 81.00% | UAVs contain specific sensors to collect information | are based on a belief approach | Protects the UAVs against the most dangerous threats |
| [23] | 2018 | 94.00% | UAV stores | SVM | Detects malicious anomalies that threaten the network |
| [24] | 2019 | 92.00% | GPS signals | artificial neural network | Detects GPS spoofing signals |
| Our approach | 2022 | 99.99% | Dataset [19] | anomaly detection | Authentication verification |

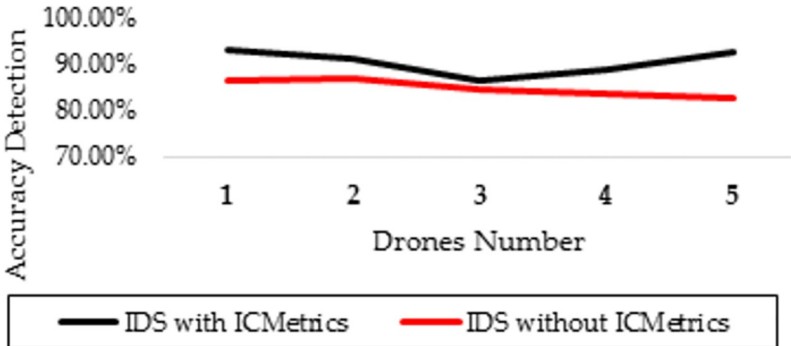

**Fig 13. Comparison between the proposed system and other IDS systems.**

following phases: simulation, data collection, feature extraction, preprocessing, training, and final testing. The deep neural network technique process plays a significant role in improving the proposed system's accuracy rate. The results of the simulation showed that the proposed detection system is accurate with respect to correctly classifying normal and abnormal behaviours and can detect malicious states and isolate them. The accuracy rates for the training and testing phases were 99.99% and 99%, respectively. In future works, the proposed system can be designed with other artificial intelligence techniques, such as support vector machines and linear discrimination analysis. In addition, the ICMetric can be extracted from ultrasound sensors equipped with a UAV.

## Supporting information

**S1 Data.**
(XLS)

## Author Contributions

**Conceptualization:** Fawaz Khaled Alarfaj.

**Data curation:** Khattab M. Ali Alheeti.

**Formal analysis:** Khattab M. Ali Alheeti.

**Investigation:** Mohammed Alreshoodi, Duaa Al Dosary.

**Methodology:** Khattab M. Ali Alheeti.

**Project administration:** Naif Almusallam.

**Resources:** Mohammed Alreshoodi, Duaa Al Dosary.

**Software:** Mohammed Alreshoodi, Naif Almusallam, Duaa Al Dosary.

**Supervision:** Khattab M. Ali Alheeti, Mohammed Alreshoodi.

**Visualization:** Fawaz Khaled Alarfaj, Naif Almusallam.

**Writing – original draft:** Naif Almusallam, Duaa Al Dosary.

**Writing – review & editing:** Naif Almusallam, Duaa Al Dosary.

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
