## [Decision Letter · Decision Letter 0]

21 Mar 2022

PONE-D-21-31928A Hybrid Security System for Drones Based on ICMetric TechnologyPLOS ONE

Dear Dr. Alheeti,

Thank you for submitting your manuscript to PLOS ONE. After careful consideration, we feel that it has merit but does not fully meet PLOS ONE’s publication criteria as it currently stands. Therefore, we invite you to submit a revised version of the manuscript that addresses the points raised during the review process.

We look forward to receiving your revised manuscript.

Kind regards,

Yongbo Li

Academic Editor

PLOS ONE

Journal Requirements:

3.We suggest you thoroughly copyedit your manuscript for language usage, spelling, and grammar. If you do not know anyone who can help you do this, you may wish to consider employing a professional scientific editing service. 

A clean copy of the edited manuscript (uploaded as the new *manuscript* file).

Reviewers' comments:

Reviewer's Responses to Questions

**Comments to the Author**

1. Is the manuscript technically sound, and do the data support the conclusions?

Reviewer #1: Partly

Reviewer #2: Partly

2. Has the statistical analysis been performed appropriately and rigorously? 

Reviewer #1: I Don't Know

Reviewer #2: Yes

3. Have the authors made all data underlying the findings in their manuscript fully available?

Reviewer #1: No

Reviewer #2: No

4. Is the manuscript presented in an intelligible fashion and written in standard English?

Reviewer #1: No

Reviewer #2: Yes

5. Review Comments to the Author

Reviewer #1: This paper can be accepted after few more works. The whole article is not well written for a research paper. The explanation is on surface and ambiguous. The term detection is mentioned throughout the article but I don t really get what kind of detection the writer addressed. The format I believe not well maintained. Some of the sentence seem has no purpose and odd. I cant really understand the problem of detection as the writer only explained in general about the issue in drone security. The writer need to polish his writing by addressing the details and sincerely explained the work done. I also cant see any data used in this paper that can help me understand the generation of iCMetric.

Reviewer #2: 1. The principle of ICMetric has been repeated several times. Similar sentences and messages are observed in Introduction Para, ICMETROC Para, ICMetric Generation and Proposed ICMetric Detection. It maybe considered to move all fundamental literature about ICEMetric in one para.

2. The Proposed ICMetric detection system lacks clarity on the specifics of the system design.

3. The Proposed DNN lacks clarity on the specifics of the algorithm.

4. Five neurons have been found to be the most optimized configuration of the hidden layers of the DNN. However, figure 3 shows 4 Neurons.

5. A Typo in line 162.

6. It maybe considered to include the details of the proposed system in Table 7 to facilitate an easier comparison.

7. The Data label of Figure 13 X Axis should be "Number of Drones".

8. The arrow is intersecting "Apply Each Time" text. The text may be moved to a side.

6. PLOS authors have the option to publish the peer review history of their article (what does this mean?). If published, this will include your full peer review and any attached files.

Reviewer #1: No

Reviewer #2: No

---

## [Decision Letter · Decision Letter 1]

21 Feb 2023

A Hybrid Security System for Drones Based on ICMetric Technology

PONE-D-21-31928R1

Dear Dr. Alheeti,

We’re pleased to inform you that your manuscript has been judged scientifically suitable for publication and will be formally accepted for publication once it meets all outstanding technical requirements.

Kind regards,

Kapil Kumar Nagwanshi, PhD

Academic Editor

PLOS ONE

Additional Editor Comments (optional):

Reviewers' comments:

Reviewer's Responses to Questions

**Comments to the Author**

1. If the authors have adequately addressed your comments raised in a previous round of review and you feel that this manuscript is now acceptable for publication, you may indicate that here to bypass the “Comments to the Author” section, enter your conflict of interest statement in the “Confidential to Editor” section, and submit your "Accept" recommendation.

Reviewer #2: All comments have been addressed

Reviewer #3: All comments have been addressed

2. Is the manuscript technically sound, and do the data support the conclusions?

Reviewer #2: Yes

Reviewer #3: Yes

3. Has the statistical analysis been performed appropriately and rigorously? 

Reviewer #2: Yes

Reviewer #3: Yes

4. Have the authors made all data underlying the findings in their manuscript fully available?

Reviewer #2: Yes

Reviewer #3: Yes

5. Is the manuscript presented in an intelligible fashion and written in standard English?

Reviewer #2: Yes

Reviewer #3: Yes

6. Review Comments to the Author

Reviewer #2: (No Response)

Reviewer #3: All comments are addressed in revised manuscript. Revised manuscript may be accepted for possible publication

7. PLOS authors have the option to publish the peer review history of their article (what does this mean?). If published, this will include your full peer review and any attached files.

Reviewer #2: **Yes: **Md Mijanur Rahman

Reviewer #3: No

---

## [Editor Report · Acceptance letter]

6 Mar 2023

PONE-D-21-31928R1 

A Hybrid Security System for Drones Based on ICMetric Technology 

Dear Dr. Alheeti:

I'm pleased to inform you that your manuscript has been deemed suitable for publication in PLOS ONE. Congratulations! Your manuscript is now with our production department. 

Kind regards, 

on behalf of

Dr. Kapil Kumar Nagwanshi 

Academic Editor

PLOS ONE